# Electrically-driven Yagi-Uda antennas for light

René Kullock[1,2]*, Maximilian Ochs[1,2], Philipp Grimm[1], Monika Emmerling[1] & Bert Hecht [1]*

Yagi-Uda antennas are a key technology for efficiently transmitting information from point to point using radio waves. Since higher frequencies allow higher bandwidths and smaller footprints, a strong incentive exists to shrink Yagi-Uda antennas down to the optical regime. Here we demonstrate electrically-driven Yagi-Uda antennas for light with wavelength-scale footprints that exhibit large directionalities with forward-to-backward ratios of up to 9.1 dB. Light generation is achieved via antenna-enhanced inelastic tunneling of electrons over the antenna feed gap. We obtain reproducible tunnel gaps by means of feedback-controlled dielectrophoresis, which precisely places single surface-passivated gold nanoparticles in the antenna gap. The resulting antennas perform equivalent to radio-frequency antennas and combined with waveguiding layers even outperform RF designs. This work paves the way for optical on-chip data communication that is not restricted by Joule heating but also for advanced light management in nanoscale sensing and metrology as well as light emitting devices.

[1] Nano-Optics and Biophotonics Group, Experimentelle Physik 5, Universität Würzburg, Am Hubland, 97074 Würzburg, Germany. [2] These authors contributed equally: René Kullock, Maximilian Ochs. *email: kullock@physik.uni-wuerzburg.de; hecht@physik.uni-wuerzburg.de

Yagi-Uda antennas consisting of a reflector, an active feed element and directors are a brilliant source of radiation as they allow locally generated electromagnetic fields to be emitted in a specific direction by means of interference effects (see Fig. 1). Owing to reciprocity, they are also very sensitive for radiation coming from a given direction and, hence, were crucial for enabling television broadcasting and had been installed on many rooftops. Miniaturizing the Yagi-Uda concept from the radio-wave to the optical regime promises two key benefits: the bandwidth dramatically grows due to the much higher frequencies and at the same time the footprint shrinks down to the nanometer scale. The resulting devices represent an efficient link between electron-based integrated computer chips and photon-based fiber networks and in particular enable on-chip optical data communication because antennas outperform subwavelength waveguides for longer distances[1], allow for multiple beam crossings, have an adaptable footprint and are not restricted by Joule heating[2,3].

Realizing optical Yagi-Uda antennas encompasses two key challenges: (i) precise fabrication of an arrangement of nanostructures and (ii) the selective driving of only one of these elements. Even though quite some effort has already been devoted to optical Yagi-Uda antennas[4]—best possible design parameters have been found for both vertical[5] as well as in-plane emitting antennas[6] and different kinds of antennas have been realized[7–12]—the main drawback of the hitherto approaches is that the light is not generated locally but bulky lab-scale setups are needed as excitation sources. Generating light locally at the nanoscale is possible by different means, for example via scanning tunneling microscopes (STMs)[13,14], carbon nanotubes[15–18], quantum dots[19]

and optical antennas[20–22]. However, obtaining directed electrically driven emission is only possible by utilizing STMs[23,24], which again involves bulky lab-scale setups, or by twisting the arms of electrically driven dipole antennas in order to break the point symmetry[25]. The latter show a limited geometrical definition and directionality only, are by design not scalable to significantly higher values and, hence, not suitable for, e.g., cross-talk free on-chip data communication. Therefore, key breakthroughs in antenna design, quality and fabrication are still needed to achieve on the nanoscale the same performance, versatility and usability as for classical radio-frequency (RF) antennas.

Here, we demonstrate the feasibility of a complex electro-optical nanosystem that consists of multiple antenna elements with precisely adjusted positions and resonances as well as a sophisticated electrical subsystem to achieve highly directed light emission via inelastic tunneling. As an example, we realize electrically driven Yagi-Uda antennas for light, that require single-crystalline connector wires[26,27], advanced focused-ion beam milling (FIB) as well as novel fabrication methods such as feedback-controlled single-particle dielectrophoresis (DEP). We experimentally show that the resulting optical antennas consisting of one reflector and three directors have unprecedented forward-to-backward (FB) ratios of up to 9.1 dB and are scalable up to 15 elements resulting in 13.2 dB. Simulations further suggest that switching to hybrid systems consisting of antennas embedded in high-index films can even outperform the characteristics of conventional Yagi-Uda antennas in the RF regime. This work opens the road to high bandwidth on-chip data communication that is not restricted by Joule heating but also for advanced light

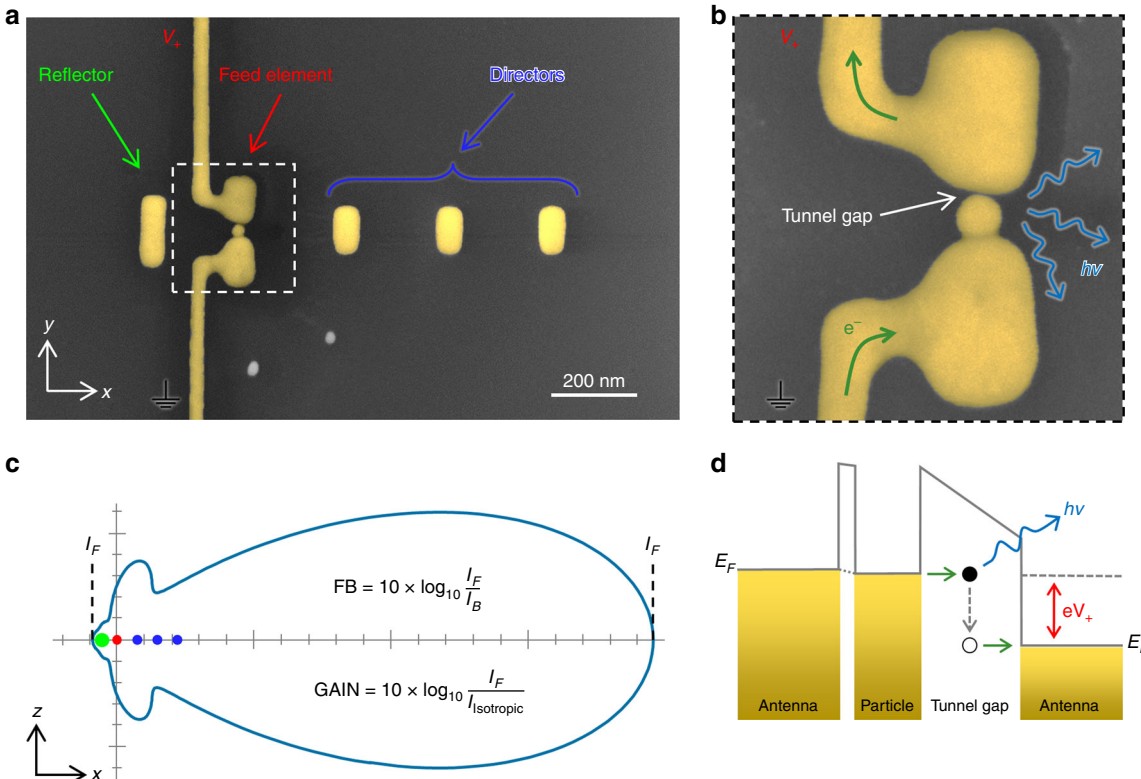

**Fig. 1 Concepts of a Yagi-Uda antenna and inelastic electron tunneling. a** SEM micrograph of a Yagi-Uda antenna containing reflector, feed element with kinked connectors and three directors on a glass substrate. **b** Zoom into the feed element highlighting the asymmetrically positioned particle creating a tunnel gap toward the top antenna arm. **c** Modeled emission characteristics of a Yagi-Uda antenna in the *xz* plane (for a homogeneous surrounding). The reflector and director elements lead to a highly unidirectional emission by generating constructive interference in the forward and destructive interference in the backward direction. The intensities in forward and backward direction can be used to define a forward-to-backward (FB) ratio and the antenna gain, see inset. **d** Schematics of the IET process.

management in nanoscale sensing and metrology as well as light-emitting devices.

## Results

**Design and fabrication of the antennas.** The electrically connected Yagi-Uda antenna systems consist of gold structures placed directly on glass in order to guarantee an undisturbed optical access via an immersion oil objective from below (cf. Fig. 2a). They are fabricated by FIB milling of chemically grown single-crystalline gold microplatelets[28] (Supplementary Note 8). The feed elements are electrically connected via FDTD-optimized kinked single-crystalline wires (see Supplementary Note 5 and Fig. 1a, b) to evaporated electrode structures that are accessed via micromanipulators (Supplementary Note 7). This arrangement provides a low-resistance electrical connection to the gap region without disturbing the optical fields[27].

For generating light, we employ inelastic electron tunneling (IET), which was first discovered in planar MIM tunnel junctions[29] in 1976 and later on studied in STM experiments[30]. When electrons tunnel over a nanoscale barrier, inelastic processes can occur in which the electrons lose energy by generating light (cf. Fig. 1d). The efficiency of IET can be strongly enhanced by a high local density of optical states (LDOS)[20–22]. IET offers distinct advantages such as the absence of any active materials resulting in a large bandwidth while offering efficiencies of up to 2%[22]. In the previous work, we employed a two-step process to implement the required 1-nm tunnel gap: first antenna structures having a ~25-nm gap were fabricated by FIB milling and then gold particles with a 1-nm thick CTAB shell were drop-casted on the sample. Suitable particles were then pushed into the antenna gaps using an atomic force microscope cantilever[20]. While successful, this approach has several drawbacks: it is problematic to push particles over longer distances, i.e. a coverage

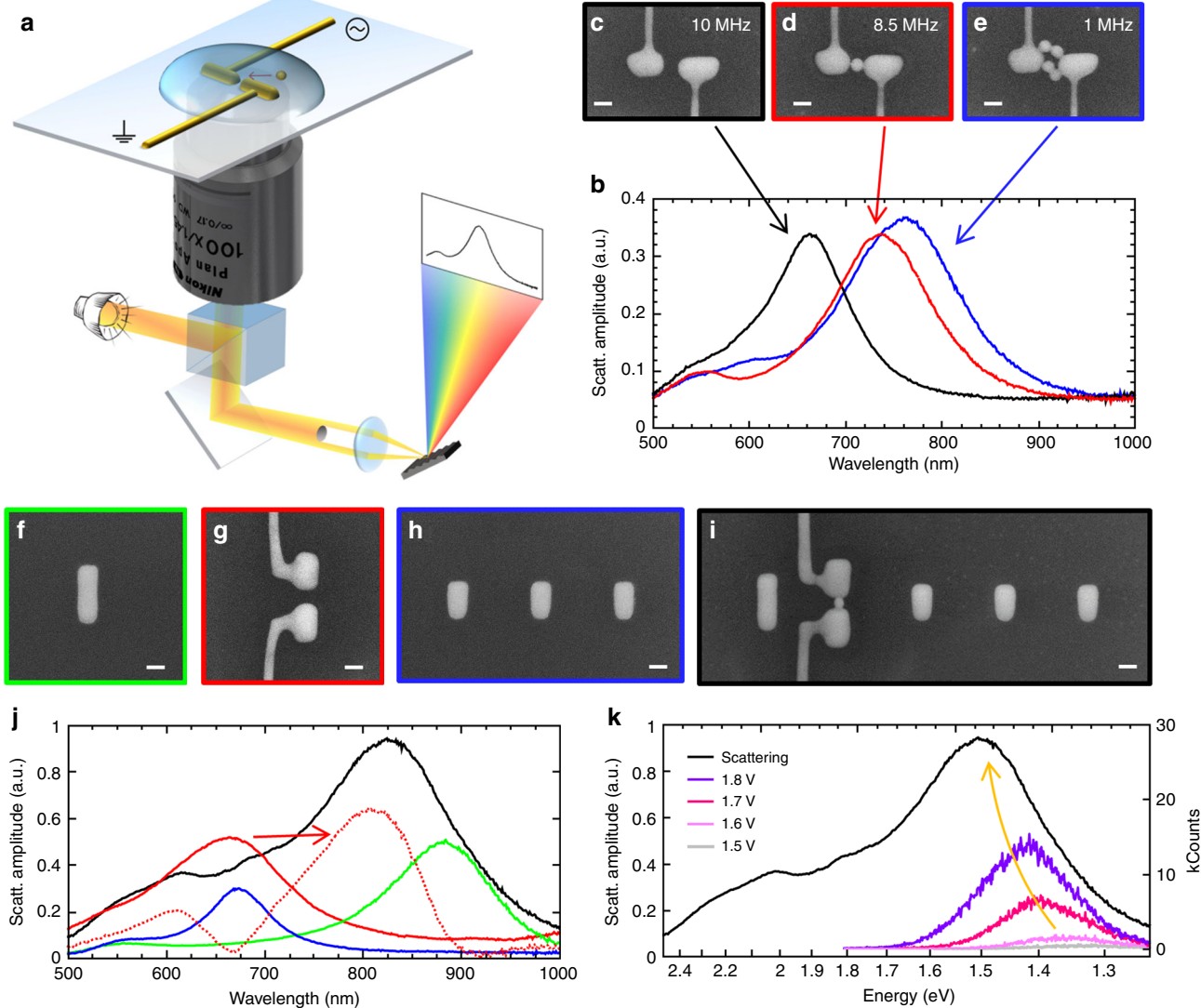

**Fig. 2 Feedback-driven dielectrophoresis and characterization of the Yagi-Uda antennas. a** Schematics of the DEP setup: while white-light dark-field scattering spectra are continuously acquired, a high-frequency AC voltage is applied. As soon as a particle is deposited in the gap, the spectrum significantly redshifts as depicted in **b** and the voltage is subsequently switched off. **c–e** Associated SEM images for various DEP frequencies showing an optimum of one-particle attraction at 8.5 MHz. **f–i** SEM images and **j** associated spectra of a reflector (green), an unloaded feed element (red), the directors (blue) as well as a completely assembled Yagi-Uda antenna (black). The dotted red line in **j** is the delta of complete antenna minus the directors and reflector and corresponds to the feed element now loaded and redshifted due to the particle inside the gap. **k** Scattering spectrum vs energy of the Yagi-Uda antenna (same as in **j**) and resulting electroluminescence (EL) spectra for various voltages. With growing voltages, the EL peak blueshifts and gets stronger as indicated by the yellow arrow (cf. ref. [7]). Note the voltage drop-off is close to zero. Scale bars, 50 nm.

of ~1 particle per square micrometer is required, the particles get easily stuck when touching the antennas at unintended locations and only a minority of the particles pushed into the antenna gap actually touch both arms and lead to functional devices. Furthermore, surplus particles have to be removed from the vicinity of the antennas to allow undisturbed optical characterization. As Yagi-Uda antennas are geometrically much more complex, this drop-and-push approach is not suitable anymore. Therefore, we introduce DEP[31,32] into the fabrication process and advance it for controlled single-particle deposition by implementing a feedback mechanism.

To perform DEP, a water droplet containing gold particles is placed on top of the antenna structures (cf. Fig. 2a). One of the two electrodes is grounded while an alternating electric field is applied to the other electrode in order to polarize the particles in solution. Depending on the voltage and frequency, particles are then attracted to regions of the highest field gradient, i.e. the feed gap. To ensure that exactly one particle is placed into the antenna gap, we continuously monitor the white-light scattering spectrum of the antenna at 10 Hz repetition rate. When a particle enters the feedgap, the spectrum strongly redshifts and the shift becomes stronger with every further particle (see Fig. 2b–e). We thus optimized the basic parameters (voltage, frequency, dilution) and achieved a success rate of single-particle deposition of 49.8% (see Supplementary Note 9).

**Optimization of the geometry**. To obtain best-possible FB ratios (see definition in Fig. 1c), the distances and sizes of the individual Yagi-Uda elements have been optimized (see Supplementary Notes 1 and 6). In principle, the reflector resonance needs to be red-shifted against the driving frequency and the director resonance blue-shifted to obtain directionality. A quasi-static dipolar interaction model[6] (Supplementary Note 3) showed that when assuming resonances of the reflector and director at 890 nm and 680 nm, respectively, and placing a reflector in 200 nm distance from the feed element, the FB ratio reaches a maximum for driving frequencies of 860 nm and a director spacing of either 200 or 330 nm. More accurate numerical Boundary Element Method (BEM) calculations[33] (Supplementary Note 4), which include retardation, higher-order interactions and an inhomogeneous surrounding (the air–glass interface), lead to a slight blue-shift of the driving frequency to 850 nm, FB ratios around 8 dB and optimal director spacing of ~130 or ~240 nm (Supplementary Fig. 6). For practical reasons, we chose the latter spacing for our experiments.

In order to fabricate such a Yagi-Uda antenna, we first studied the individual elements separately and matched their optical properties (resonance position and width) to our models by slight adjustments of the fabricated geometry. In Fig. 2f–j, SEM images and associated scattering spectra of the final elements as well as a fully assembled antenna are shown. As intended, the resonance positions of the reflector and directors occur at 890 nm and 680 nm, respectively, while the feed element and the whole Yagi-Uda antenna are resonant around 800 nm. (Note, the driving frequency will be on the red side of the resonance and because the feed element red-shifts differently depending on the detailed particle location (cf. ref.[20]), its spectrum is estimated by subtracting the passive elements from the Yagi-Uda in Fig. 2j.) All geometrical parameters are listed in the Supplementary Table 1.

**Optoelectronic characterization**. Electroluminescence was measured by applying a DC voltage of up to 1.8 V and collecting the emitted light via a high-NA objective. The corresponding results in Fig. 2k exhibit an emission peak which blue-shifts and

increases in amplitude with increasing voltage. Previous experiments showed that the high LDOS in the antenna gap is responsible for an enhanced inelastic tunneling rate—i.e. the emission peak—and that the blue-shift as well as the amplitude increase can be explained by a quantum-shot noise model[20]. In order to prevent destruction of the antenna, we limited the applied voltage to 1.8 V, which results in an emission maximum around 870 nm that is close to ideal for driving the Yagi-Uda antenna. Furthermore, the voltage drop-off between applied voltage and maximum emitted photon energy in eV is close to zero indicating only a single tunneling barrier. This is in line with high-resolution SEM images acquired after all optical measurements were completed (cf. Fig. 1b).

In order to experimentally estimate the FB ratio, we recorded the emission pattern of the electroluminescence by back-focal plane imaging for various antennas and evaluated them with the common 'pixel' method used by Curto et al.[8] and the more accurate 'areal' method introduced by Gurunarayanan et al.[25]. See Supplementary Note 2 for the definitions and error discussions.

First, as a reference, we investigated a dipole antenna (Fig. 3a) and observed a FB ratio of 1.5 ± 1.4 dB and 0.1 ± 0.2 dB with both pixel and areal method, respectively, which is close to the theoretical expected 0 dB. Figure 3 also depicts the results for three experimental Yagi-Uda antennas. The first Yagi-Uda antenna is the one discussed in Fig. 2 and shows a directionality of 6.6 dB or 5.3 dB, respectively. This is a larger value than the maximum of 6 dB (pixel method) measured by Curto et al. for optical driven Yagi-Uda antennas as well as the maximal 5 dB (areal method) obtained by Gurunarayanan et al. with their twisted dipolar antenna approach. We were able to fabricate several antennas with similar or better performance and the two

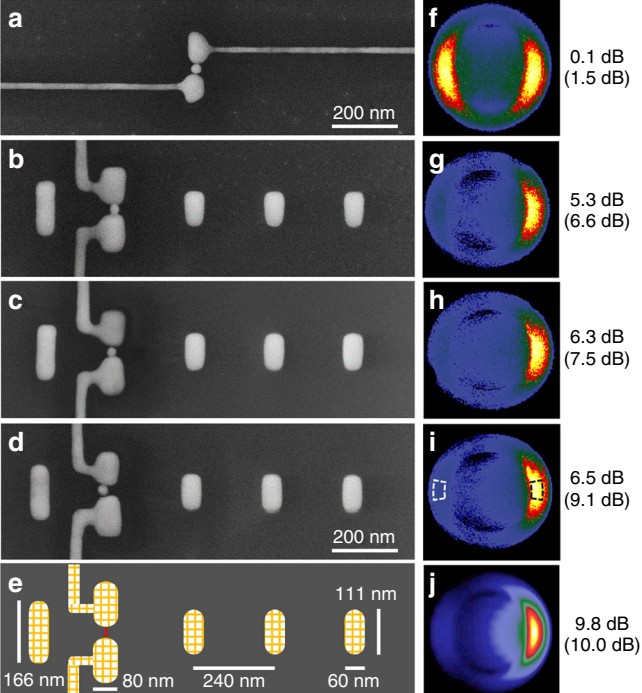

**Fig. 3 Comparison of dipole and Yagi-Uda antennas. a–e** SEM micrographs of one dipole as well as three Yagi-Uda antennas and a sketch of the FDTD model. **f–j** Corresponding emission patterns. While the dipole antenna has a balanced emission, the Yagi-Uda antennas show a high directionality to the right (forward direction) and nearly no emission to the left (backward direction). The adjacent numbers give the resulting FB ratios determined by the areal (pixel) method. The integration areas are indicated in **i**.

remaining antennas in Fig. 3 show FB ratios of up to 9.1/6.5 dB. These results exceed the values of hitherto published optical antennas and, hence, highlight the potential of electrically driven Yagi-Uda antennas for light. Variations between the individual antennas can be traced back to slight deviations in antenna geometry and particle placement. Furthermore, the results are qualitatively and quantitatively very close to the numerical results from FDTD calculations (see Supplementary Note 5 and Table 1) shown in Fig. 3j indicating a limit of this design around 10 dB.

**Limits of Yagi-Uda antennas in the optical regime.** In order to further improve the FB ratio, simply more directors can be added to the arrangement analogous to the RF regime[34]. We have fabricated antennas with up to 15 elements (see Supplementary Note 10) and show exemplary emission characteristics in Fig. 4c. Even though these antennas emitted light when applying a DC voltage, the FB ratios were surprisingly low with only 3.1 dB (pixel method) and 1.9 dB (areal method), respectively, which are below the smallest FB-ratio seen in three-director antennas. Overall, we found the tendency that with increasing number of directors, the FB ratio decreases. The reason for this counterintuitive behavior is the asymmetric air–glass dielectric

surrounding and the subsequent refraction of light into the higher-index substrate. This means every additional director, which is placed a step further away from the source, is reached by less field and, hence, can contribute less to a directed emission (Supplementary Note 11). Furthermore, the additional path length the light needs to travel from the source via the more distant directors to the detector sitting below the substrate results in a phase lag and, therefore, a slight destructive interference at the detector, i.e. less signal. With every further director, this destructive interference increases which explains the low FB values for the 15-element antenna. A droplet of immersion oil on top of the structures would easily circumvent these problems as it provides a symmetric dielectric surrounding but it also prevents a straight forward measurement of the emission pattern—the main lobe being pushed beyond the acceptance angle of even high-NA objectives. Such experiments would require advanced detection schemes and are therefore beyond the scope of the present study.

Nevertheless, emission in a symmetric surrounding can be simulated and compared to conventional RF Yagi-Uda antennas to judge their performance. We therefore simulated an antenna embedded in a homogenous $n = 1.52$ surrounding, adjusted the geometry to match the new dielectric environment and plotted the resulting $xz$ emission pattern in Fig. 4b. As expected, the shape of the pattern is now symmetric and features a much higher FB ratio of up to 13.2 dB at 870 nm. This corresponds to an antenna forward gain of 11.7 dBi. (The antenna forward gain as defined in Fig. 1c and Supplementary Note 11 is the Figure of merit in RF antenna technology; unfortunately, it is in contrast to the FB ratio not easily accessible in nano-optical experiments.) In Fig. 4a, this forward gain is plotted as a function of the number of directors for an optical antenna (emission wavelength 870 nm) and also for a conventional RF Yagi-Uda (500 MHz, stainless steel, see Supplementary Note 11) using standard algorithms[35,36]. In both cases, the forward gain starts at relatively moderate values for a small number of directors, initially increases strongly with increasing number of directors but then flattens out and reaches a plateau around 12.0 dBi and 15.1 dBi, respectively. This means that the directionality characteristic of this optical Yagi-Uda antenna is similar to an RF antenna.

Since gold is known for its high absorption losses in the visible regime, we ran simulations under less lossy circumstances on the one hand by increasing the wavelength to the telecom regime (1.55 μm) where intraband absorptions do no longer contribute, and on the other hand by artificially reducing the imaginary part of the dielectric function to 10% of its nominal value. This already leads to an increase of the maximum gain to 12.8 dBi and 13.9 dBi, respectively (see Supplementary Fig. 18). By substituting gold with the less lossy silver, the RF values can even be reached as depicted in Fig. 4a. This proves that optical Yagi-Uda antennas perform analogous to their RF counterparts and, therefore, concepts of RF antenna theory should be easily transferable to optical Yagi-Uda designs to further improve or adapt their performance.

We also note that optical Yagi-Uda antennas are even outperforming RF antennas for small numbers of directors. The reason is that in the RF case, the connecting wires are considered to be infinitesimally small and, hence, negligible. In the optical case, the connectors inevitably have a finite size and therefore act as additional passive elements. Therefore, we included them in our models and optimized their location to enhance the directivity (Supplementary Note 5), which becomes especially apparent for small numbers of directors.

Finally, for optical fields, it is possible to mold the flow of light by designing a dielectric index landscape. For radiowaves, this ability is very restricted due to the lack of suitable materials. Here, we consider embedding the Yagi-Uda antenna in a thin film with

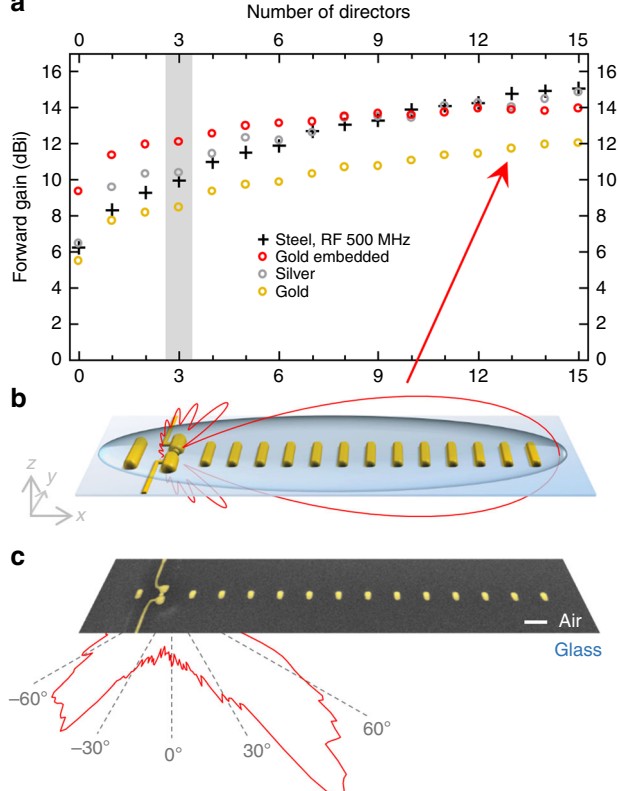

**Fig. 4 Limits of an optical Yagi-Uda antenna. a** Calculated forward gain of a conventional stainless-steel Yagi-Uda antenna in the RF regime (500 MHz) for a varying number of directors (black crosses). Its counterparts in the optical regime (870 nm) for gold and silver antennas (golden and silver circles) as well as a hybrid system consisting of gold antennas embedded in a 300-nm Al$_2$O$_3$ layer (red circles). The case for three directors is highlighted (gray area). **b** Perspective sketch of the optical antenna with 13 directors in an assumed homogeneous surrounding ($n = 1.52$) and the calculated $xz$ emission characteristics superimposed. **c** SEM image of an actually built Yagi-Uda antenna with 13 directors and superimposed measured $xz$ emission characteristics. The bending to the bottom is due to the air–glass interface. Scale bars, 200 nm.

high refractive index in order to confine the emitted light to a 2D waveguide mode. By embedding antennas into a 300-nm thick $Al_2O_3$ layer and adapting again the geometry, we were able to increase the forward gain drastically by up to 3.6 dBi. This means that for nearly any number of directors, a waveguide-coupled optical Yagi-Uda antenna outperforms the RF stainless steel and pure optical silver antennas—see red circles in Fig. 4a. The increase in performance is especially apparent for low numbers of directors (e.g. to 12.1 dBi for three directors) and also particularly interesting because it opens the road toward highly directive optical antennas with very small footprint.

## Discussion

Highly-directive low-footprint optical Yagi-Uda antennas are specifically promising for on-chip data communication applications where they can act both as sender and receiver of data with an ultrahigh bandwidth. The directive connectors and the thin dielectric cover are inherently necessary for the antennas in order to be integrated into computer chips and at the same time are very beneficial for their performance. Furthermore, the resulting communication schemes would also allow transistors from one computer chip to be directly linked via optical networks to transistors on other chips, which reduces latencies and allows novel computational concepts. Therefore, we believe that optical Yagi-Uda antennas will play a major role in future computational devices. Additionally, the method of placing nanoparticles in specific positions by means of DEP could also be utilized to deposit functional or optically active particles such as nanodiamonds or quantum dots into the gaps paving the way for quantum-optical application such as single-photon sources or quantum-sensing devices. Finally, the presented manufacturing toolbox provides a powerful platform to easily implement more elaborate antenna configurations such as log antennas or antennas with a parabolic reflector as well as novel devices such as electrically driven plasmonic waveguides.

## Methods

**Fabrication**. In a wet-chemical process, single-crystalline gold flakes are grown on glass coverslips (24 mm × 24 mm #1.5 Menzel) via reduction of chloroauric acid $HAuCl_4$ in ethylene-glycol[28]. Electrode structures are prepared by optical lithography as well as electron beam physical vapor deposition and consist of a 20-nm chromium adhesion layer and 80-nm gold layer. The Flakes are then transferred[20] to the electrodes and structured by FIB milling (Helios Nanolab, FEI Company, Oregon, USA) using an acceleration voltage of 30 kV and a beam current of 9.7 pA in order to obtain the nanoantennas. For the DEP, the sample is mounted on a Nikon TE2000-U inverted microscope upgraded with a Mad City Labs Inc. Nano-LPS200 nanopositioning stage. A diluted solution of gold nanoparticles (A11C-30-CTAB-1, Nanopartz, Loveland, USA) is drop-casted onto the antennas, individual antennas are brought into focus and white-light scattering spectra (see below) are continuously acquired. The antennas are contacted via micromanipulators and an AC voltage is applied using a Standford Research Instruments DS 345 function generator until the spectra red-shift indicating particle deposition. After several depositions are performed, the sample is unmounted and washed with ethanol and water in order to prevent further accidental particle deposition.

**Optical characterization**. A stabilized halogen lamp (Thorlabs SLS201L/M) coupled into a 20-μm multi-mode fiber serves as an excitation source for the dark-field white-light scattering measurement. The light is outcoupled via a reflective collimator (Thorlabs RC08), sent through a 300-μm pinhole and focused via a 500-mm lens into the back focal plane of an oil-immersion microscope objective (Plan-Apochromat, 100×, NA = 1.45, Nikon) in order to illuminate the sample with a parallel light beam. A circular beam block is mounted in the detection path such that only scattered light from the structures can pass and direct reflections are blocked. The scattered light is analyzed by a spectrometer (Shamrock 303i, 80 lines/mm blazing at 870 nm) in combination with an electron-multiplied charge-coupled device (EMCCD, iXon A-DU897-DC-BVF, Andor).

**Electrical characterization**. Current–voltage characteristics are acquired by a source measure unit (SMU, Keithley 2636B, Keithley Instruments Inc., Cleveland, USA), which is wired to probe heads (DPP220, Cascade Microtech, Beaverton,

USA) via triax cabling. Copper-beryllium probe needles (Semprex Corp., Cambell USA) are mounted to the heads and used to contact the electrodes on the samples.

**Electro-optical measurements**. A sample is mounted on the Nikon TE2000-U microscope, the internal beam splitter as well as the circular beam block are removed and a voltage is applied to a specific antenna using the electrical setup from above. The resulting electroluminescence is recorded by the same Shamrock spectrometer and Andor camera from above. To allow correlated data recording, the SMU and EMCCD camera are synchronized via a LabVIEW program. Typical integration times of the camera are 100 ms for spectra and 30 s for emission pattern. For recording the latter, a 100-mm Betrand lens is introduced into the beam path[20] and the grating is replaced by a mirror.

## Data availability

The data that support the findings of this study are available from the corresponding authors upon reasonable request.

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

## Acknowledgements

The VW-Foundation (Grant 93437) and the German Research Foundation (HE 5618/4-1) are acknowledged for financial support.

## Author contributions

R.K. conceived and set up the experiments. M.E. fabricated the electrode structures and transferred the platelets. M.O. and R.K. optimized the FIB milling. M.O. conducted the DEP, WL and EL measurements. P.G. setup the analytical model and performed the BEM and FDTD simulations. R.K., M.O., P.G. and B.H. analyzed the data and co-wrote the manuscript. B.H. supervised the work.

## Competing interests

The authors declare no competing interests.
