## [Peer Review File · Nature Communications]

Reviewers' comments:

Reviewer #1 (Remarks to the Author):

The manuscript convincingly demonstrates electrically-driven directional emission of an optical Yagi-Uda nanoantenna using a nanopositioned tunnel junction as the light source. In simulations, the optical Yagi Uda even outperforms RF designs thanks to an optimized geometry of the driving electrodes acting as passive optical elements and embedding in a homogeneous dielectric environment. The current work bridges the existing gap between optical antennas and radio frequency antennas, because the difficulty of electrical driving of nanoantennas significantly limits their application in information and communication technologies.

The present work is a nanofabrication and characterization tour de force, building on the demonstration of electrically-driven nanoantennas published by some of the authors in Nature Photonics in 2015 and taking it to much more complex directional antenna designs. In terms of relevance, I do not expect the manuscript to directly "influence thinking in the field" (one of the aspects that reviewers are asked to comment about), as it is quantitative or implementation rather than conceptual work. Having said this, it clearly pushes the limits of this incipient technology and brings prospects of application to nanoantennas, so in my opinion, it perfectly fits Nature Communications.

As the technical correctness of the results and analysis seems clear, my comments focus on aspects that can increase the clarity or reproducibility of the manuscript:

- Identify explicitly at the end of the introduction differences or improvements over reference 25, which showed electrically-driven directional antennas in 2017.
- Describe the reproducibility of positioning one single nanoparticle at the gap using DEP. It is a statistical process, so do all devices end up with a single particle deterministically? The current description is succinct: "We thus optimized the basic parameters (voltage, frequency, dilution) to obtain single particle deposition only."
- Enumerate explicitly for clarity the "key breakthroughs still needed to achieve on the nanoscale the same performance, versatility and usability as for RF antennas," or rephrase the paragraph.
- Indicate explicitly the reason for the impossibility of applying AFM positioning for Yagi-Udas, as mentioned in the sentence "As Yagi-Uda antennas geometries are much more complex, the drop & push approach cannot be applied anymore."
- Point to the supplementary information where relevant. For example, the pixel and areal methods are defined there but this fact is not mentioned in the relevant paragraph in the main text. The same occurs for the supplementary section about FDTD optimization of the contact geometry, that is not clearly referenced in the main text.

Note: at some points, informal/inaccurate writing, typos or possible overstatements decrease the rigorous quality of the text. Examples: "a very brilliant source", "had been installed on many rooftops", "QDots", "loose" instead of "lose", "we believe that optical Yagi-Uda antennas will play a major role in future computational devices", "these results vastly exceed the values of hitherto published optical antennas"

Reviewer #2 (Remarks to the Author):

Kullock et al. present an electrically-driven Yagi-Uda antenna that operates at optical frequencies.

Photons are generated by inelastic electron tunneling at a small gap in the feed element. The small gap (of the order of 1 nm) is obtained by placing a nanoparticle in between the feed element by dielectrophoresis. The electrical contacts of the feed element are designed to comply with the working principles of a Yagi-Uda antenna. Reflector and director elements are obtained by detuning nanoparticle resonances with respect to the feed element. The resulting device is investigated using back-focal plane measurement of the emission pattern and the antenna directivity is inferred by two different approaches. Results are compared to literature values both in the optical and radio-frequency range. A significant amount of work is also devoted to optimization aspects. Applications for on-chip optical data transfer are discussed.

This is a piece of fine experimental work in nanophotonics. The main motivation/novelty is the electrical excitation of the feed element "the main drawback of the hitherto approaches is that the light is not generated locally but bulky lab-scale setups are needed as excitation sources". The conceptual/technological advancement sounds interesting for the optical-antennas community, in my opinion, since the proposed electrical-excitation scheme and the fabrication methods are not readily scalable. There is little comparison with other directional antenna designs, not based on resonant Yagi-Uda antennas, which could also be more favorable for electrical excitation. For example, in <https://doi.org/10.1364/OME.7.001634> realistic optical antenna designs with a gain of 18.4 dBi using only one director layer are presented. See also the experimental results in <https://www.nature.com/articles/lsa2016245> and <https://arxiv.org/pdf/1905.03363.pdf>.

I have the following technical comments:

- In the feedback driven electrophoresis it would be interesting to see the spectral difference between one and two nanoparticles, since this would clarify the sensitivity of the feedback.
- In Fig. 2k (Scattering spectrum vs energy of the Yagi-Uda antenna and resulting electroluminescence (EL) spectra for various voltages) it is not clear what the white-light spectrum refers to. Maybe to the scattering spectrum under white-light excitation? I haven't found this mentioned in the text.
- "Fig. 3 show FB ratios of up to 9.1/6.5 dB. These results vastly exceed the values of hitherto published optical antennas and, hence, highlight the potential of electrically-driven Yagi-Uda antennas for light". It would be helpful to indicate the error in the determination of the forward-backward (FB) ratio, since the difference between area and pixel methods is not negligible and the backward emission seems close to the noise level. Probably, an estimate based on a fit of the radiation pattern would be less affected by noise. Is the theoretical FB ratio estimated like the experimental one?
- What are the exact parameters of the radio-frequency Yagi-Uda antenna chosen for comparison with the optical ones?

Mario Agio

Reviewer #3 (Remarks to the Author):

This is a very good and solid paper. It is really well-written, and it is honest about the techniques and the design methods used. There is no hype or extra self-citations.

Yagi-Uda antennas have been studied for a number of years, but the demonstrations in the optical regime are limited. The authors employ a compact electrical driving, and demonstrate electrically-driven Yagi-Uda antennas for light with wavelength-scale footprints that exhibit large directionalities. The resulting antennas perform equivalent to RF antennas and combined with wave guiding layers even outperform RF designs.

I am not an experts in the experimental techniques these authors used but I do not have concerns based on how everything is explained in the text. I believe that it would be suitable for Nature

Communications.

Reviewer #4 (Remarks to the Author):

Referee report: Electrically-driven Yagi-Uda antennas for light

The manuscript is devoted to the fabrication of Yagi-Uda antennas, which are electrically driven using an inelastic tunneling process. In order to obtain such an antenna several steps are needed which are very well and detailed explained in the manuscript and/or supplementary information. First, a detailed modelling is done to define the required dimensions of the antenna structure. Secondly, an advanced sample fabrication process was developed combining focused ion beam milling to obtain a high quality Au antenna structure and feedback-controlled dielectrophoresis was used to position a single Au nanoparticle inside the gap. Thirdly, a strong directional emission is observed for various Yagi-Uda antennas. In a final part, the authors explore the possibility to enhance the directionality by increasing the number of directors (up to 15). However, this did not lead to the expected increased directionality. The observed trend was explained via the asymmetric air-glass dielectric surrounding which causes refraction of light into the higher index substrate and reduces the impact of the additional directors farther away from the source. Although the authors state that they cannot symmetrize the environment in their experimental setup, the additional modelling shows that embedding the structure into a thin dielectric cover (e.g. 300 nm AL₂O₃) increases the forward gain strongly. These results clearly show the potential of electrically driven yagi-Uda antenna's for on-chip communication.

To summarize my viewpoint on the manuscript: the data is well presented and without doubt a clear forward gain is achieved. It builds on the expertise of the group related to electrically driven optical antennas, using a Au nanoparticle, and now couples them to a well established Yagi-Uda antenna design. From this perspective it does not bring a radically new idea. As stated in their introduction, it adds to reference [1] and [2] the compact electrical driving. While, the work done in Ref [25] of the presented manuscript already achieved electrically driven directionality of the radiation using a v-shaped antenna. However, to my opinion it still represents an important step forward in combining electrical and optical signals on chip. As such, I believe it has enough added value to warrant publication in nature communications. Still I have some questions/suggestions related to the manuscript, which are given below:

- 1) Although a fascinating approach is used to create the nanogap needed for inelastic tunneling. It would be interesting to know how the authors would introduce a feeding element which is compatible with conventional fabrication process.
- 2) For example in reference [2] the authors used thermally evaporated Au, while in this work the authors use single crystalline gold. Is this really a strict requirement for the overall success of their approach?
- 3) As tunneling is by default quantum I would remove this adjective.
- 4) In the supplementary materials the authors explore how the contacts can already induce some non-zero FB ratio. However, it is clear from the SEM pictures that also the location of the nanoparticle is non-symmetric (See figure S1.5). Can the authors comment on the impact of the Au position on the overall FB ratio?
- 5) What is meant with "man-machine" interfaces in the introduction?

6) The statement: "As Yagi-Uda antennas geometries are much more complex, the drop & push approach cannot be applied anymore" is unclear. At the end, the nanoparticle has to be placed in between the nanogap. The only thing which changes is the surrounding of the nanogap. Can the authors specify this need for another approach?

7) The authors use two methods to determine the FB ratio. The method of Curto is perhaps more prone to fluctuations, while the method of Gurunaryanan underestimates a bit the directionality by averaging over a broader area. Keeping this in mind and, as the dipole antenna shows in both cases a non-zero value, it would certainly be good to add errors on these numbers.

8) In the modelling, figure 4, even another definition is used from antenna technology. The so-called forward gain. In order to strengthen the link between experiment and theory it would be better to use the same definition throughout the manuscript.

7) Figure 3. As done in the supplementary information (figure S1.5). It would be nice if the authors could show the simulated emission patterns for the antenna design as well. Already to see how well the model captures the emission pattern.

8) If one looks to the inelastic tunneling model (figure 1d), the voltage drop is across one gap only. Is this model always valid for all devices.? So do you always obtain the voltage drop off between applied voltage and max photon energy to be zero? Because if I compare the position of the Au nanoparticle it certainly is not identical for all devices.

9) What is the success rate of the dielectrophoresis? I can imagine that the observation of a change in the resonance does not necessarily mean you will have a good tunnel barrier?

Response to the referees' comments

We would like to thank all reviewers for their fast review.

Reviewer #1

The manuscript convincingly demonstrates electrically-driven directional emission of an optical Yagi-Uda nanoantenna using a nanopositioned tunnel junction as the light source. In simulations, the optical Yagi Uda even outperforms RF designs thanks to an optimized geometry of the driving electrodes acting as passive optical elements and embedding in a homogeneous dielectric environment. The current work bridges the existing gap between optical antennas and radio frequency antennas, because the difficulty of electrical driving of nanoantennas significantly limits their application in information and communication technologies.

The present work is a nanofabrication and characterization tour de force, building on the demonstration of electrically-driven nanoantennas published by some of the authors in Nature Photonics in 2015 and taking it to much more complex directional antenna designs. In terms of relevance, I do not expect the manuscript to directly "influence thinking in the field" (one of the aspects that reviewers are asked to comment about), as it is quantitative or implementation rather than conceptual work. Having said this, it clearly pushes the limits of this incipient technology and brings prospects of application to nanoantennas, so in my opinion, it perfectly fits Nature Communications.

As the technical correctness of the results and analysis seems clear, my comments focus on aspects that can increase the clarity or reproducibility of the manuscript:

- Identify explicitly at the end of the introduction differences or improvements over reference 25, which showed electrically-driven directional antennas in 2017.

➔ In the revised manuscript we point out at the end of the introduction the differences/improvements over ref 25 more clearly.

- Describe the reproducibility of positioning one single nanoparticle at the gap using DEP. It is a statistical process, so do all devices end up with a single particle deterministically? The current description is succinct: "We thus optimized the basic parameters (voltage, frequency, dilution) to obtain single particle deposition only."

➔ In the revised manuscript (page 5) and SI (pages 16/17) we added information about the statistics of the DEP process.

- Enumerate explicitly for clarity the "key breakthroughs still needed to achieve on the nanoscale the same performance, versatility and usability as for RF antennas," or rephrase the paragraph.

➔ In the revised manuscript (page 3) we state what breakthroughs are needed.

- Indicate explicitly the reason for the impossibility of applying AFM positioning for Yagi-Udas, as mentioned in the sentence "As Yagi-Uda antennas geometries are much more complex, the drop & push approach cannot be applied anymore."

➔ In the revised manuscript (page 5) we now lay out the difficulties of the drop & push approach such that the reader can understand why it is not suitable for Yagi-Uda antennas.

- Point to the supplementary information where relevant. For example, the pixel and areal methods are defined there but this fact is not mentioned in the relevant paragraph in the main text. The same occurs for the supplementary section about FDTD optimization of the contact geometry, that is not clearly referenced in the main text.

→ Each subsection in the SI is now numbered and the manuscript was updated to more clearly point to relevant information in the SI.

Note: at some points, informal/inaccurate writing, typos or possible overstatements decrease the rigorous quality of the text. Examples: “a very brilliant source”, “had been installed on many rooftops”, “QDots”, “loose” instead of “lose”, “we believe that optical Yagi-Uda antennas will play a major role in future computational devices”, “these results vastly exceed the values of hitherto published optical antennas”

→ Wikipedia: “The greater the brilliance, the more photons of a given wavelength and direction are concentrated on a spot per unit of time.” So, we think “Yagi-Uda antennas ... are a very brilliant source of radiation” is a suitable wording.

→ “...were crucial for enabling television broadcasting and had been installed on many rooftops.” We agree that “had been installed on many rooftops” could easily be left out. However, readers not familiar with Yagi-Uda antennas might be happy to get a hint what Yagi-Uda antennas actually are.

→ The performance of computer chips is mainly limited by the available data bandwidth. Therefore, modern processors have a complex data-cache hierarchy, long instruction pipelines where data are prefetched, execution branch prediction to already pull new data, simultaneous multithreading and so on. Typical values for on-chip cache latencies are 4-5 cycles (~1 ns) for the L1, 12-14 cycles (~4 ns) for the L2 and 36-68 cycles (>9 ns) for the L3 cache. Light beams can travel within 0.1 ns a distance of 2 cm ($n=1.5$) and thanks to the bosonic nature can cross each other without disturbances. Hence, accessing the L2 and L3 caches via optical connections could reduce their latencies significantly and improve the overall computational performance. Therefore we state: “we believe that optical Yagi-Uda antennas will play a major role in future computational devices”

→ We revised the manuscript to reflect the other suggestions.

Reviewer #2

Kullock et al. present an electrically-driven Yagi-Uda antenna that operates at optical frequencies. Photons are generated by inelastic electron tunneling at a small gap in the feed element. The small gap (of the order of 1 nm) is obtained by placing a nanoparticle in between the feed element by dielectrophoresis. The electrical contacts of the feed element are designed to comply with the working principles of a Yagi-Uda antenna. Reflector and director elements are obtained by detuning nanoparticle resonances with respect to the feed element. The resulting device is investigated using back-focal plane measurement of the emission pattern and the antenna directivity is inferred by two different approaches. Results are compared to literature values both in the optical and radio-frequency range. A significant amount of work is also devoted to optimization aspects. Applications for on-chip optical data transfer are discussed.

This is a piece of fine experimental work in nanophotonics. The main motivation/novelty is the electrical excitation of the feed element “the main drawback of the hitherto approaches is that the light is not generated locally but bulky lab-scale setups are needed as excitation sources”. The conceptual/technological advancement sounds interesting for the optical-antennas community, in my opinion, since the proposed electrical-excitation scheme and the fabrication methods are not readily scalable. There is little comparison with other directional antenna designs, not based on resonant Yagi-Uda antennas, which could also be more favorable for electrical excitation. For example, in <https://doi.org/10.1364/OME.7.001634> realistic optical antenna designs with a gain of 18.4 dBi using only one director layer are presented. See also the experimental results in <https://www.nature.com/articles/lsa2016245> and <https://arxiv.org/pdf/1905.03363.pdf>.

→ Even though vertical emitting Yagi-Uda antenna are not the focus of this manuscript, we included a differentiation and cited the two peer-reviewed papers from the referee in the revised version.

I have the following technical comments:

- In the feedback driven electrophoresis it would be interesting to see the spectral difference between one and two nanoparticles, since this would clarify the sensitivity of the feedback.

→ The situation for two particle is complicated as they sometimes accidentally align such that both particles touch both antenna arms. In that case, the spectrum is not distinguishable from the case of a single particle inside the gap. However, in other cases, the two particles align in-line and a larger shift is visible. Nevertheless, the added Fig. S2.8 in the revised SI shows that it is 4-5 times more likely to deposit a single particle instead of two.

- In Fig. 2k (Scattering spectrum vs energy of the Yagi-Uda antenna and resulting electroluminescence (EL) spectra for various voltages) it is not clear what the white-light spectrum refers to. Maybe to the scattering spectrum under white-light excitation? I haven't found this mentioned in the text.

→ In the revised manuscript, we make it clear that it is the same scattering spectrum as in 2j.

- “Fig. 3 show FB ratios of up to 9.1/6.5 dB. These results vastly exceed the values of hitherto published optical antennas and, hence, highlight the potential of electrically-driven Yagi-Uda antennas for light”. It would be helpful to indicate the error in the determination of the forward-backward (FB) ratio, since the difference between area and pixel methods is not negligible and the backward emission seems close to the noise level. Probably, an estimate based on a fit of the radiation pattern would be less affected by noise. Is the theoretical FB ratio estimated like the experimental one?

→ In the revised SI (Supplementary Note S1.1) the errors of the FB ratios are now discussed. The theoretical FB ratios are estimated like the experimental ones. This is made clear by the new Fig 3j in the revised manuscript.

- What are the exact parameters of the radio-frequency Yagi-Uda antenna chosen for comparison with the optical ones?

➔ The exact parameters are given in the SI. We updated the manuscript and the SI (Supplementary Note S3.3) to make that clearer.

Reviewer #3

This is a very good and solid paper. It is really well-written, and it is honest about the techniques and the design methods used. There is no hype or extra self-citations. (thank you)

Yagi-Uda antennas have been studied for a number of years, but the demonstrations in the optical regime are limited. The authors employ a compact electrical driving, and demonstrate electrically-driven Yagi-Uda antennas for light with wavelength-scale footprints that exhibit large directionalities. The resulting antennas perform equivalent to RF antennas and combined with wave guiding layers even outperform RF designs.

I am not an experts in the experimental techniques these authors used but I do not have concerns based on how everything is explained in the text. I believe that it would be suitable for Nature Communications.

➔ No action needed.

Reviewer #4

The manuscript is devoted to the fabrication of Yagi-Uda antennas, which are electrically driven using an inelastic tunneling process. In order to obtain such an antenna several steps are needed which are very well and detailed explained in the manuscript and/or supplementary information. First, a detailed modelling is done to define the required dimensions of the antenna structure. Secondly, an advanced sample fabrication process was developed combining focused ion beam milling to obtain a high quality Au antenna structure and feedback-controlled dielectrophoresis was used to position a single Au nanoparticle inside the gap. Thirdly, a strong directional emission is observed for various Yagi-Uda antennas. In a final part, the authors explore the possibility to enhance the directionality by increasing the number of directors (up to 15). However, this did not lead to the expected increased directionality. The observed trend was explained via the asymmetric air-glass dielectric surrounding which causes refraction of light into the higher index substrate and reduces the impact of the additional directors farther away from the source. Although the authors state that they cannot symmetrize the environment in their experimental setup, the additional modelling shows that embedding the structure into a thin dielectric cover (e.g. 300 nm AL₂O₃) increases the forward gain strongly. These results clear show the potential of electrically driven yagi-Uda antenna's for on-chip communication.

To summarize my viewpoint on the manuscript: the data is well presented and without doubt a clear forward gain is achieved. It builds on the expertise of the group related to electrically driven optical antennas, using a Au nanoparticle, and now couples them to a well established Yagi-Uda antenna design. From this perspective it doesnot bring a radically new idea. As stated in their introduction, it adds to reference [1] and [2] the compact electrical driving. While, the work done in Ref [25] of the presented manuscript already achieved electrically driven directionality of the radiation using a v-shaped antenna. However, to my opinion it still represents an important step forward in combing electrical and optical signals on chip. As such, I believe it has enough added value to warrant publication in nature communications. Still I have some questions/suggestions related to the manuscript, which are given below:

1) Although a fascinating approach is used to create the nanogap needed for inelastic tunneling. It would be interesting to know how the authors would introduce a feeding element which is compatible with conventional fabrication process.

- As we point out in the revised SI the success rate for single particle deposition is close to 50%. This is a very good value for these proof-of-principle experiments and the majority of failed attempts is due to clustering of particles already in solution. (We had to reorder the particle solution 3 times from the manufacture as the clustering got stronger with the age of the solution.) We think that on the industrial scale the yield could therefore be improved significantly.
- Furthermore, wet-chemical processes are needed in conventional fabrication processes anyways and standard probing systems could be utilized for conducting the DEP. As a feedback mechanism one could also use optical reflection spectra or – once optimized – direct DC measurements over the tunnel barriers.

2) For example in reference [2] the authors used thermally evaporated Au, while in this work the authors use single crystalline gold. Is this really a strict requirement for the overall success of their approach?

→ We use single-crystalline gold, because we need a high accuracy for structuring the individual elements (especially the kinks close to the feed elements) but also to obtain a low resistance of the connectors. When evaporating gold the films usually grow in a Volmer-Weber mode, which leads to grains that are disadvantageous for both the accuracy and resistance. However, a recent publication shows that this can be prevented by cooling the substrate down to liquid nitrogen temperatures during growth. Combining this with state-of-the-art EUV lithography might be enough for a large-scale fabrication of electrically-connected optical Yagi-Uda antennas.

→ <https://doi.org/10.1021/acsp Photonics.9b00907>

3) As tunneling is by default quantum I would remove this adjective.

→ We removed this adjective in the revised manuscript.

4) In the supplementary materials the authors explore how the contacts can already induce some non-zero FB ratio. However, it is clear from the SEM pictures that also the location of the nanoparticle is non-symmetric (See figure S1.5). Can the authors comment on the impact of the Au position on the overall FB ratio?

→ The location of the nanoparticle has no significance influence on the spectrum and on the emission pattern. A detailed analysis can be found here:

→ <https://doi.org/10.1117/12.2289647>

5) What is meant with “man-machine” interfaces in the introduction?

→ We removed “man-machine interfaces” in the revised manuscript.

6) The statement: “As Yagi-Uda antennas geometries are much more complex, the drop & push approach cannot be applied anymore” is unclear. At the end, the nanoparticle has to be placed in between the nanogap. The only thing which changes is the surrounding of the nanogap. Can the authors specify this need for another approach?

→ We revised the manuscript (page 5) to better explain the difficulties with the drop & push approach.

7) The authors use two methods to determine the FB ratio. The method of Curto is perhaps more prone to fluctuations, while the method of Gurunaryanan underestimates a bit the directionality by averaging over a broader area. Keeping this in mind and, as the dipole antenna shows in both cases a non-zero value, it would certainly be good to add errors on these numbers.

→ In the revised SI (Supplementary Note S1.1) the errors of the FB ratios are now discussed.

8) In the modelling, figure 4, even another definition is used from antenna technology. The so-called forward gain. In order to strengthen the link between experiment and theory it would be better to use the same definition throughout the manuscript.

→ In antenna theory/technology the antenna gain is the figure of merit for comparing antennas and it uses an isolate (dipole) emitter as reference. Unfortunately, in nano-optical experiments one cannot simply isolate the emitter – especially when it is electrically driven. Therefore, the FB ratio is commonly used for comparing optical antennas in literature and we have to use our theoretical simulations to link between these two worlds.

→ We made a comment on that in the revised manuscript on page 10.

7) Figure 3. As done in the supplementary information (figure S1.5). It would be nice if the authors could show the simulated emission patterns for the antenna design as well. Already to see how well the model captures the emission pattern.

→ We updated Fig. 3 with the simulated emission patterns and would like to thank the reviewer for this suggestion.

8) If one looks to the inelastic tunneling model (figure 1d), the voltage drop is across one gap only. Is this model always valid for all devices.? So do you always obtain the voltage drop off between applied voltage and max photon energy to be zero? Because if I compare the position of the Au nanoparticle it certainly is not identical for all devices.

→ Our experience with hundreds of devices is that the particle nearly always sits asymmetrically, indicating only one significant tunneling barrier. However, sometimes a voltage drop-off is observed which indicates multiple tunneling junctions. This can usually be overcome by applying a high “activation” voltage which presumably breaks down one of the two barriers and leads to light emission.

9) What is the success rate of the dielectrophoresis? I can imagine that the observation of a change in the resonance does not necessarily mean you will have a good tunnel barrier?

→ In the revised manuscript (page 5) and SI (pages 16/17) we added information about the statistics of the DEP process. The success rate is with >48% one order of magnitude higher than with the drop & push approach (for dipole antennas).

→ The placement of single particles inside the gaps (using DEP) usually results in light emission based on inelastic electron tunneling. We recorded reasonable electroluminescence spectra in 78 % of these cases.

REVIEWERS' COMMENTS:

Reviewer #1 (Remarks to the Author):

The authors have addressed my questions satisfactorily. I recommend this article for publication in Nature Communications.

Reviewer #2 (Remarks to the Author):

The revisions have improved the manuscript and the authors have properly addressed the reviewers comments. I do not have any further request and recommend the manuscript for publication in Nat. Comm.

Mario Agio

Reviewer #4 (Remarks to the Author):

To my opinion, the authors addressed my remarks regarding the manuscript and I recommend it for publication in nature communications.